# Study on the Remediation of Pyrene-Contaminated Soil with Surfactants and their Mechanisms

**Liang Shen** [1,2,*], **Yifang Liu** [2], **Jiabao Gong** [2] **and Erle Qiao** [2]

1   State Key Laboratory of Mining Response and Disaster Prevention and Control in Deep Coal Mines, Huainan 232001, China
2   College of Material Science and Engineering, Anhui University of Science and Technology, Huainan 232001, China
*   Correspondence: shen654520@126.com

**Abstract:** Soil is the main aggregation site of polycyclic aromatic hydrocarbons and an important pathway of migration to other media. In this paper, the adsorption behavior of pyrene and seven different types of surfactants on kaolinite surfaces was studied by molecular dynamics simulation and desorption testing. The molecular dynamics simulation results showed that pyrene was more easily adsorbed on the 001 (-) side of kaolinite. SDBS, SDS, TW80, and TX-100 had strong interactions with pyrene, encapsulating pyrene molecules in aggregates. However, when the concentration of surfactant was too high, the desorption of pyrene molecules on a kaolinite surface will be inhibited. The desorption of pyrene molecules will be inhibited in the presence of BS-12, TW80, and TX-100, while the desorption process can be promoted by using CTAC, DDBAC, SDBS, and SDS as soil remediation agents. The removal rate of pyrene gradually increased with the increase of SDS dosage, while for SDBS, the removal rate showed a trend of first increasing and then decreasing. When the concentration of SDS was 0.014 mol/L, the elution rate of pyrene reached 72.86%. The molecular dynamics simulation results were similar to the desorption test results, verifying the reliability of molecular dynamics simulation. The research results provide theoretical support for the selection of surfactants in the remediation process of pyrene-contaminated soil.

**Keywords:** soil remediation; polycyclic aromatic hydrocarbons; surfactants; molecular simulation

## 1. Introduction

As a clay mineral, kaolinite is a white-layered silicate product produced by natural weathering of feldspar and silicate minerals. It has good plasticity, insulation, acid, and alkali resistance and excellent adhesion. It is widely used in ceramics, papermaking, textiles, rubber, environmental protection, biological medicine, and refractory and other industries. Kaolinite is a very important mineral with large reserves around the world. However, kaolinite is also a gangue mineral that needs to be removed in many industries. For example, in iron ore flotation, coal flotation, and slime precipitation, kaolinite is an unfavorable mineral, which should be removed.

Polycyclic aromatic hydrocarbons (PAHs) are a kind of aromatic hydrocarbon containing two or more benzene rings, which are produced in the process of incomplete combustion of coal, oil, wood, and other organic substances. Due to their carcinogenicity, genetic toxicity, and teratogenic effects, they pose a toxic risk to the ecological environment and public health, and have been listed as a priority pollutant in many countries [1–4].

PAHs exist widely in nature and easily accumulate in soil and other media. At the same time, they can also enter the human body through the food chain, causing harm [5]. Polycyclic aromatic hydrocarbons (PAHs) are semivolatile and have different volatility according to different molecular structures. For PAHs with weaker volatility, such as pyrene, it can be adsorbed in the soil for a long time, causing continuous pollution to the environmental atmosphere, water, and soil. Since soil is one of the main media for

polycyclic aromatic hydrocarbons' migration and enrichment, it is necessary to study how to effectively reduce or control the content of polycyclic aromatic hydrocarbons in soil. Wei et al. [4]. concluded through experiments that for pure clay minerals and soil, the desorption effect of single or mixed surfactants on pyrene is consistent. In order to further identify the effects of clay minerals in soils on the desorption of pyrene using single or mixed anionic–nonionic surfactants, two clay minerals (montmorillonite and kaolin) were selected. The results showed that for pure kaolin, the effectiveness of desorption was consistent with the desorption observed from soil. Yang et al. [6]'s analysis concluded that in underground environments with typically low organic carbon content, polycyclic aromatic hydrocarbons adsorb a significant portion of the mineral surface. Soil organic matter is the major sorbent for PAHs in surface horizons. In the subsurface environment, where the content of the organic carbon is usually low, sorption of PAHs to a mineral surface may be of greater importance. If a sample is obtained from a soil layer 1 m below the surface with 0.378% soil organic matter, it can be concluded that clay minerals absorb a large proportion of PAHs.

Due to the hydrophobicity and slow desorption rate of polycyclic aromatic hydrocarbons (PAHs), traditional physical remediation methods are difficult to use to effectively remove PAHs from soil [7–10]. Research has shown that using surfactants to treat contaminated soil can effectively increase the desorption of PAHs [10–12]. At present, the main-stream research method is to evaluate the enhancing effect of surfactants in the remediation of PAH-contaminated soil through adsorption/desorption tests, and to analyze the macroscopic interaction mechanism of surfactants and PAHs in soil through various experimental methods such as high-performance liquid chromatography, X-ray diffraction (XRD), surface tension, and spectrophotometry [4,10,12–14]. Wei et al. [4] studied the enhanced remediation (SER) performance of single and mixed anionic–nonionic surfactants in artificial pyrene-contaminated soil using various methods such as XRD, UV spectrophotometry, and high-performance liquid chromatography. Yang et al. [8] studied the effect of a mixed solution of nonionic surfactants (t-octylphenoxypolyethoxyethanol, TX-100) and anionic surfactants (sodium dodecylbenzenesulfonate, SDBS) on the desorption capacity of phenanthrene in contaminated soil through batch experiments with different proportions of mixed surfactants. Chong et al. [12] studied the desorption effect of four surfactants on PAHs in soil of abandoned manufacturing natural gas plants with different levels of pollution using spectrophotometry and high-performance liquid chromatography.

The molecular mechanics method originated in 1970, based on the calculation method of classical mechanics. This calculation method is based on the principle of Born–Oppenheimer approximation, ignoring the process of electronic motion, taking the energy of the system as a function of the position of the atomic nucleus, and calculating various properties of molecules or systems through molecular force fields. Through molecular dynamics simulation, dynamic and thermodynamic statistical information of various systems and characteristics can be obtained, which is a widely used method for studying complex systems. Compared with the calculation method of quantum chemistry, molecular dynamics simulation does not consider the movement of electrons, but takes atoms as the smallest unit, so it can deal with larger systems. Its unique MD constraint technology and SMART structure optimization method combine the advantages of steepest descent method, conjugate gradient method, and Newton's method. Therefore, molecular dynamics simulation has the advantage of less calculation, and is suitable for simulation of large systems.

At present, molecular dynamics (MD) simulation is widely applied in the fields of biopharmaceuticals and material preparation [15,16], but there is still relatively little research on the micro interactions between surfactants, polycyclic aromatic hydrocarbons, and soil using molecular dynamics simulation. Researchers mainly use molecular simulation technology to study the micro adsorption process of pollutants in soil. Due to the large proportion of clay minerals in soil, people often use clay minerals to represent the inorganic matter in soil for research. Chen et al. [17] studied the adsorption behavior of naphthalene on clay minerals montmorillonite and kaolinite through the MD method. The results

showed that the electrostatic effect of montmorillonite was greater than that of kaolinite, so montmorillonite has a higher adsorption potential for naphthalene. Wu et al. [18] used the MD method to study the interaction process of asphalt, resin, and aromatics on the surface of quartz. The results showed that with increasing temperature, the adsorption of asphalt and aromatics did not change significantly, while the adsorption of resin became more compact. Wu et al. [19] studied the adsorption process of antibiotics on montmorillonite based on adsorption kinetics and molecular simulation, and the results showed that there was a competitive adsorption behavior between tetracycline and ciprofloxacin. Chen et al. [20] studied the thermal desorption mechanism of n-dodecane in unsaturated clay through TGA testing, a multi-component kinetic model, and MD simulation. The results showed that water and n-dodecane would have competitive adsorption on the surface of montmorillonite, but had little effect on the n-dodecane adsorbed on the surface of kaolinite.

However, the above research lacks a microscopic analysis of the interaction between surfactants, polycyclic aromatic hydrocarbons, and soil, and cannot provide accurate information on how surfactants interact with polycyclic aromatic hydrocarbons and affect their adsorption process on soil. Understanding their interactions is an important aspect of studying the remediation of polycyclic aromatic hydrocarbon-contaminated soil; therefore, further research is needed. In this study, MD simulation and desorption tests were used to study the interaction between different kinds of surfactants, pyrene, and kaolinite. Based on MD simulation, the adsorption model of surfactant and pyrene on the surface of kaolinite was constructed, and then the desorption test and MD simulation were compared and analyzed to reveal the adsorption mechanism.

## 2. Materials and Methods

### 2.1. Material Samples and Reagents

Reagents

The reagents used in the experiment, such as pyrene, tramadol X-100 (TX-100), sodium dodecyl sulfate (SDS), hexadecyltrimethylammonium chloride (CTAC), dodecyldimethylbenzylammonium chloride (DDBAC), lauryl betaine (BS-12), sodium dodecylbenzene sulfonate (SDBS), Tween 80 (TW 80), dichloromethane, methanol, etc., were all purchased from Shanghai Aladdin Reagent Co., Ltd. (Shanghai, China). The surfactant was prepared as a solution with a concentration of 0.1 mol/L. Kaolinite was sourced from Jinyan kaolinite Processing Factory, Huaibei City, Anhui Province. The abbreviation and chemical formula of surfactants are shown in Table 1. The molecular structure of all surfactants is shown in Figure 1. The chemical composition analysis results of kaolinite are shown in Table 2. It can be seen from Table 2 that the purity of the kaolinite samples was high.

**Table 1.** Chemical formulas and abbreviations of the surfactants.

| Collector | Chemical Formulas | Abbreviations |
|---|---|---|
| Pyrene | $C_{16}H_{10}$ | - |
| Tramadol X-100 | $C_{14}H_{22}O(C_2H_4O)_n$ | TX-100 |
| Sodium dodecyl sulfate | $C_{12}H_{25}SO_3Na$ | SDS |
| Hexadecyltrimethylammonium chloride | $C_{19}H_{42}ClN$ | CTAC |
| Dodecyldimethylbenzylammonium chloride | $C_{21}H_{38}ClN$ | DDBAC |
| Lauryl betaine | $C_{16}H_{33}NO_2$ | BS-12 |
| Sodium dodecylbenzene sulfonate | $C_{18}H_{29}NaO_3S$ | SDBS |
| Tween 80 | $C_{24}H_{44}O_6(C_2H_4O)_n$ | TW 80 |

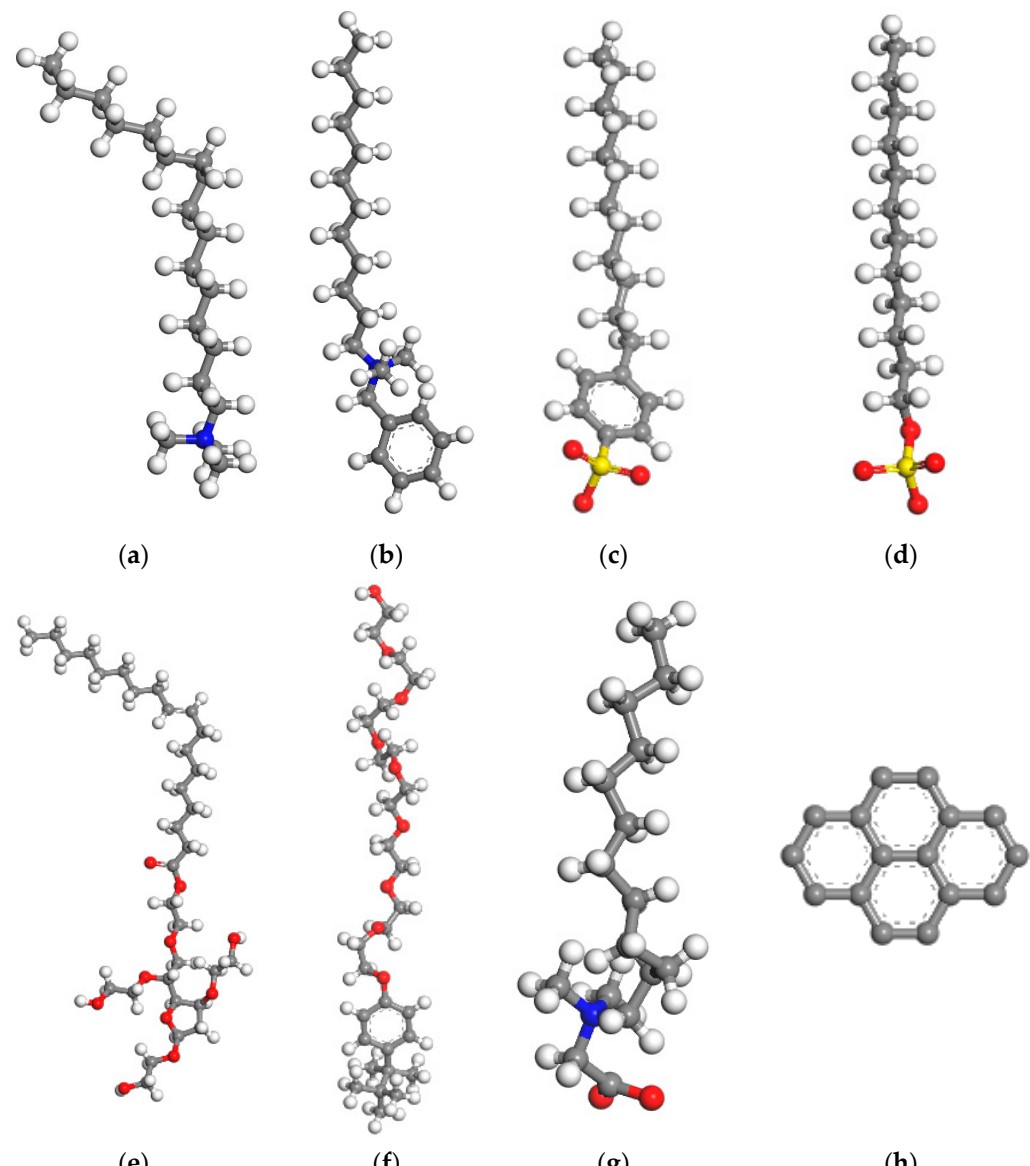

**Figure 1.** Equilibrium configuration of pyrene adsorption on (001) and 001 (-) surfaces of kaolinite. (**a**) CTAC; (**b**) DDBAC; (**c**) SDBS; (**d**) SDS; (**e**) TW80; (**f**) TX-100; (**g**) BS-12; (**h**) pyrene.

**Table 2.** Chemical components of kaolinite samples.

| SiO$_2$ | Al$_2$O$_3$ | Fe$_2$O$_3$ | MgO | CaO | Na$_2$O | K$_2$O | TiO$_2$ | MnO | Loss |
|---|---|---|---|---|---|---|---|---|---|
| 49.625 | 34.221 | 1.135 | 0.036 | 0.131 | 0.101 | 0.081 | 0.625 | 0.015 | 14.03 |

### 2.2. Preparation of Pyrene-Contaminated Samples

Kaolinite samples were crushed to 125 μm, and then dried at room temperature (25–30 °C) for a week in darkness. Then, 0.2 g of pyrene was dissolved in methanol solution, and uniformly mixed in 500 g of kaolinite to prepare the contaminated sample. The samples were placed in a stainless steel basin for 7 days, and shaken 10 times a day to improve the uniformity of pyrene pollution. The content of pyrene in the final kaolinite was 0.385 g/kg. In each desorption test, the contaminated sample was mixed evenly and a multi-point sampling method was adopted.

### 2.3. Desorption Test

We mixed 9 g of contaminated kaolinite and distilled water to prepare a 150 mL solution, and then different doses (5 mL, 10 mL, 15 mL, 20 mL, 25 mL) of surfactant solution were added into the beaker. Then, a stirrer was placed in the beaker and conditioned for 10 min at 2400 rpm to obtain the treated sample solution. After filtration, the contaminated sample was naturally air dried at room temperature (25 °C). Pyrene was extracted from the sample into dichloromethane solution using ultrasonic extraction method. We used 100 mL dichloromethane solution in the extracting process, and then 1 mL of dichloromethane solution was taken for drying. Lastly, 10 mL of methanol was added, and a UV spectrophotometer (UNESCO UV-3802) was used to determine the content of pyrene.

### 2.4. Molecular Dynamics Simulation

#### 2.4.1. Adsorption Model

Kaolinite belongs to the group of layered silicate minerals, which are mainly connected by hydrogen bonds between layers. When kaolinite is crushed, 001 surface and 001 (-) surface are mainly generated (Figure 2). In this paper, the molecular surface models of kaolinite on 001 and 001 (-) sides were obtained by using the kaolinite cell in MS software and cleaving along the 001 side, and a $3 \times 5 \times 1$ supercell surface model was constructed. Then, water, pyrene, surfactant molecules, and charge balance ions were added into the simulation system to build a pyrene kaolinite solution system, pyrene surfactant solution system, and kaolinite pyrene surfactant solution system. To avoid interactions caused by periodic boundary conditions, an 80 Å thick vacuum slab was added in all systems. Before assessing MD (molecular dynamics), energy minimization was adopted to remove abnormal van der Waals effects. Then, geometry optimization in the forcite module was used to minimize energy and obtain the initial adsorption models for each system. The number of water molecules, surfactants, and pyrene molecules in each simulation system is shown Table 3.

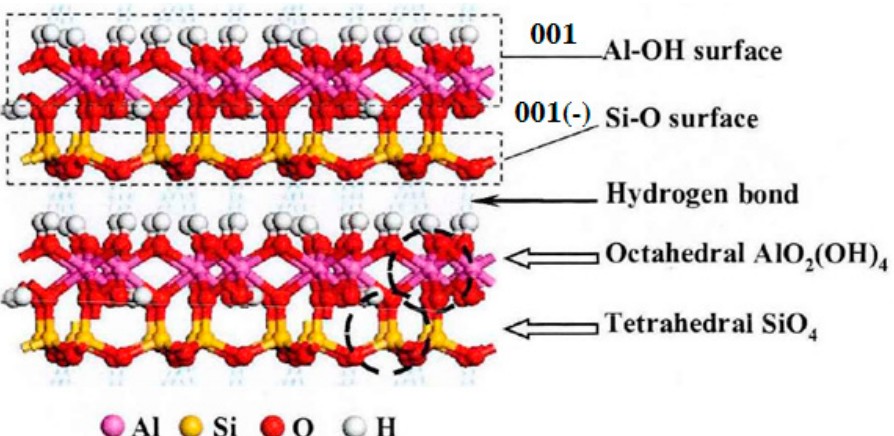

**Figure 2.** 001 and 001 (-) surfaces of kaolinite.

**Table 3.** Components of each system.

| Systems | Components | Size/(Å $\times$ Å $\times$ Å) |
|---|---|---|
| 1 | 1000 water + kaolinite + 5 pyrene | 26 $\times$ 26 $\times$ 45 |
| 2 | 3000 water + 9 surfactants + 5 pyrene | 45 $\times$ 45 $\times$ 45 |
| 3 | 1000 water + kaolinite + 5 pyrene + 9 surfactants | 26 $\times$ 26 $\times$ 45 |
| 4 | 1000 water + kaolinite + 5 pyrene + 5 surfactants | 26 $\times$ 26 $\times$ 45 |

#### 2.4.2. Simulation Method

The Forcite module was used to simulate the molecular dynamics of the adsorption of surfactants and pyrene on the surface of kaolinite. A polymer consistent force field

(PCFF) was applied in the simulation process. The Ewald method was used for long-term electrostatic interaction, and the atom-based method was used for van der Waals interaction, with a cutoff radius of 12.5 Å. The Nosé temperature control method was selected, under the NVT ensemble, the time step was set to 1.0 fs, and the total simulation time was 1000 ps. During all the simulations, the kaolinite surface was fixed. All the molecular dynamic simulations were performed with Forcite modules in the Materials Studio 2017 software developed by Accelrys Inc., San Diego, CA, USA.

## 3. Results and Discussion

### *3.1. Molecular Dynamics Simulation Results and Molecular Structure*

#### 3.1.1. Adsorption Equilibrium Configuration

The final configuration of pyrene adsorption on 001 and 001 (-) sides of kaolinite is shown in Figure 3. It can be seen from Figure 1 that pyrene was more easily adsorbed on the 001 (-) side of kaolinite. Wu et al. [5] also found that the organic pollutant m-Xylene is more likely to be adsorbed on kaolinite 001 (-) in molecular dynamics simulation. The number of weak hydrogen bonds produced by kaolinite adsorption of BTX on the 001 (-) surface was significantly higher than that on the 001 surface.

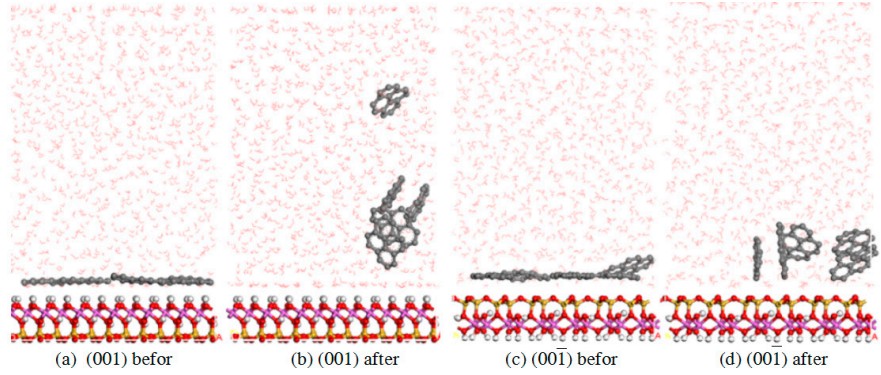

| (a) (001) befor | (b) (001) after | (c) (00$\bar{1}$) befor | (d) (00$\bar{1}$) after |

**Figure 3.** Equilibrium configuration of pyrene adsorption on (001) and 001 (-) surfaces of kaolinite.

The simulation results of surfactants and pyrene molecules in aqueous solution are shown in Figure 4. From Figure 4, it can be seen that among the seven selected surfactants, SDBS, SDS, TW80, and TX-100 all generated strong interactions with pyrene, encapsulating pyrene molecules in aggregates. Therefore, the solubilization effect of surfactants can be used to remove pyrene molecules from water.

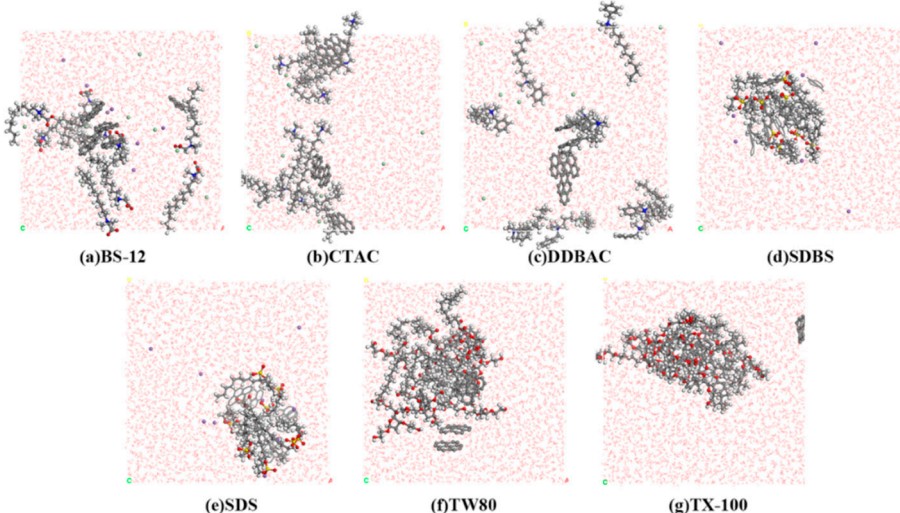

| (a)BS-12 | (b)CTAC | (c)DDBAC | (d)SDBS |
| (e)SDS | (f)TW80 | (g)TX-100 |

**Figure 4.** Simulation results of surfactants and pyrene molecules in aqueous solution.

Figures 5 and 6 showed the molecular dynamics simulation equilibrium results when the number of surfactant molecules was 5 and 9, respectively. Comparing Figures 5 and 6, it can be found that when the molecular number of the collector was 9, the pyrene molecule was always close to the surface of kaolinite, which is not conducive to the elution of pyrene molecules. When the molecular number of the collector was 5, the pyrene molecules were far away from the surface of kaolinite, and then agglomerated with the surfactant to achieve the removal of pyrene molecules. Especially with cationic surfactants such as CTAC and DDBAC, when their concentrations were high, they were adsorbed on the surface of kaolinite to form a hydrophobic layer, improving the surface hydrophobicity of kaolinite, thus inhibiting the desorption of pyrene molecules.

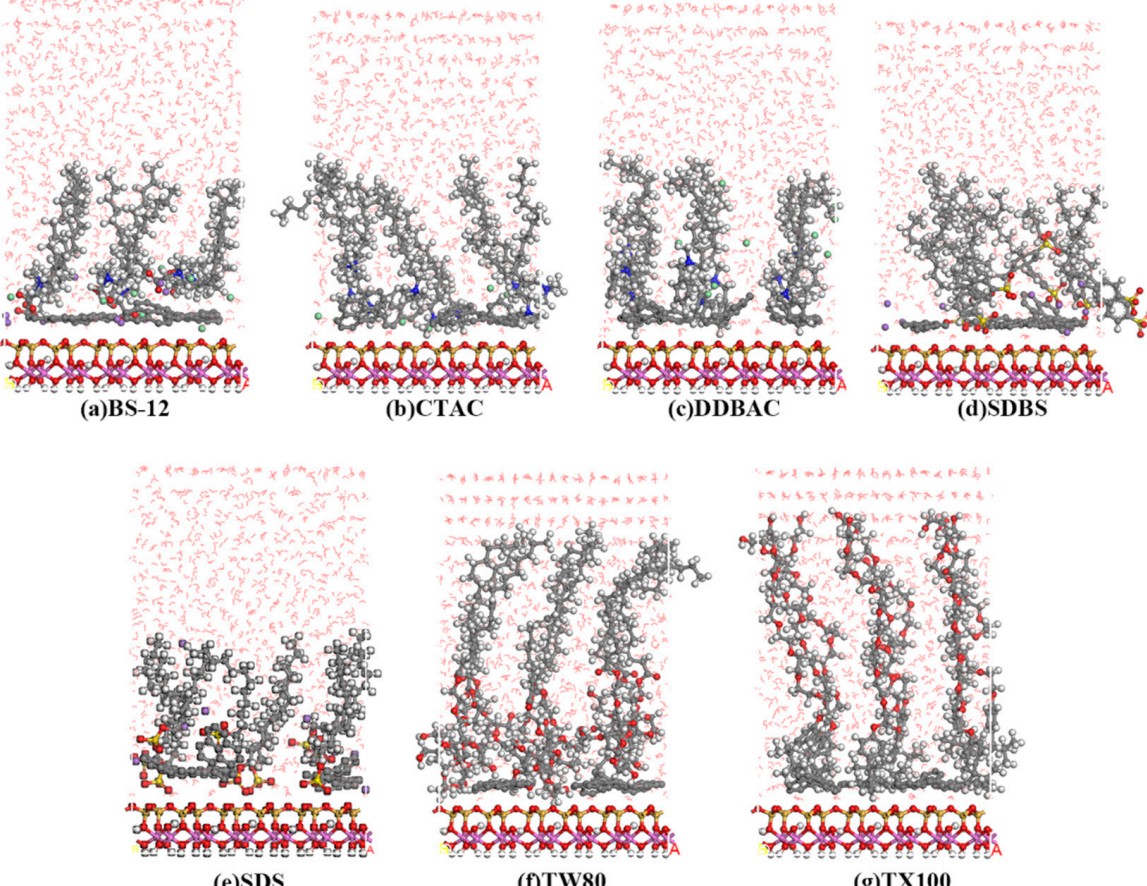

**Figure 5.** Simulation results of surfactants and pyrene molecules in aqueous solution.

### 3.1.2. Relative Concentration Distribution

The structure of the interface can be characterized by the relative density distribution, and the distribution of different atoms along the vertical direction of the interface can be calculated by the concentration distribution function in the Forcite module. The principle of the concentration distribution function is based on the mass transfer principle in molecular dynamics simulation. Mass transfer refers to the molecular movement caused by the concentration difference between molecules in different regions. The molecular movement will make the concentration tend to be uniform until the equilibrium state is reached. The concentration profile function can calculate the concentration gradient between different regions, which is the molecular motion caused by the concentration difference between different regions. The concentration gradient reflects the driving force of molecular motion, and the larger the concentration gradient, the faster the molecular motion. The concentration profile function can also calculate the concentration distribution at different time points. At the same time, the number or mass of molecules in different regions can be calculated

according to the direction and region specified to obtain the concentration distribution, and the diffusion, adsorption, reaction, equilibrium, and other processes of molecules in the system can be displayed by drawing the curve of concentration changes with position or time. We can observe the concentration changes of different components in materials in three-dimensional space. This function can help to understand the transport phenomena and phase behavior of material.

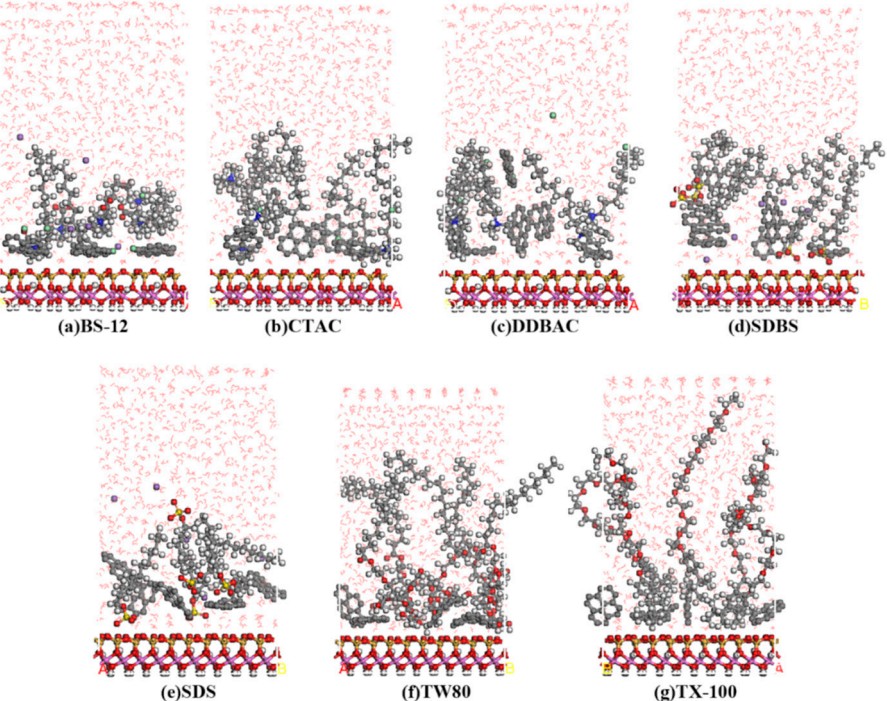

**Figure 6.** Simulation results of surfactants and pyrene molecules in aqueous solution.

In order to analyze the specific position of pyrene after desorption from kaolinite surfaces under the action of different surfactants, the relative concentration distribution of pyrene along the Z axis in each system was calculated, and is given in Figures 7 and 8. It can be seen from Figure 7 that, under the condition of high concentration of surfactant, the pyrene molecules were not far from the surface of kaolinite, but tended to be close. It can be found from Figure 8 that when the concentration of surfactant decreased, different types of surfactants had different effects on the adsorption behavior of pyrene molecules on kaolinite surfaces. The presence of BS-12, TW80, and TX-100 inhibited the desorption of pyrene molecules, while CTAC, DDBAC, SDBS, and SDS promoted the desorption of pyrene molecules.

It can be seen from the adsorption equilibrium configuration and relative concentration distribution that different concentrations of surfactants played different roles in the removal rate of pyrene on different surfaces of kaolinite. A high concentration of surfactant had an inhibitory effect on the desorption of pyrene. A large number of surfactants formed a higher stacking density on the surface of kaolinite, creating a more hydrophobic environment, thus preventing the desorption of pyrene, which is consistent with previous research [5,21]. From Figures 5 and 7, it can be seen that under low concentrations of surfactants, only the amphoteric surfactant BS-12 had a strong inhibitory effect on pyrene desorption.

Amphoteric surfactants have positive and negative polarities, which lead to stronger adsorption between pyrene and kaolinite, thus inhibiting the desorption of pyrene. When a cationic surfactant is adsorbed on the surface of kaolinite, it will compete with pyrene molecules for adsorption, thus promoting the desorption of pyrene. Relatively low concentrations of anionic surfactants are more likely to bind with pyrene and hydrophobically modify it, thereby promoting its desorption; nonionic surfactants do not ionize in water, so

they are generally adsorbed on the charged solid surface. As can be seen from Figure 1, both TW 80 and TX-100 contain a benzene ring or cycloalkane. When they exist in relatively low concentration, first the benzene ring structure in the nonionic surfactant and the pyrene on the surface of kaolinite are adsorbed by π-π stacking, and then the benzene ring structure in the nonionic surfactant generates weak hydrogen bonds with the surface of kaolinite at the same time, thus generating co-adsorption and inhibiting the desorption of pyrene. In addition, because the specific surface area of kaolinite is certain, the binding energy of pyrene molecule alone is not as large as the binding energy of a pyrene nonionic surfactant, and some pyrene molecules will be competitive adsorbed, leading to certain desorption.

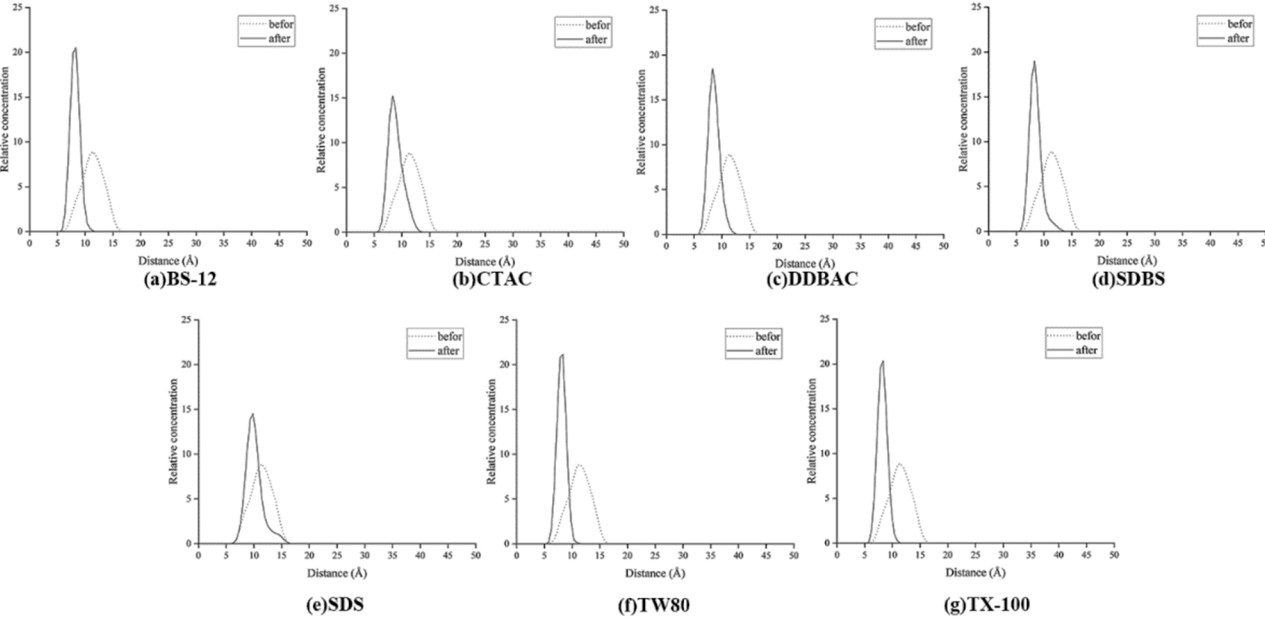

**Figure 7.** Concentration distribution of pyrene adsorbed on kaolinite surface under a high concentration of surfactant.

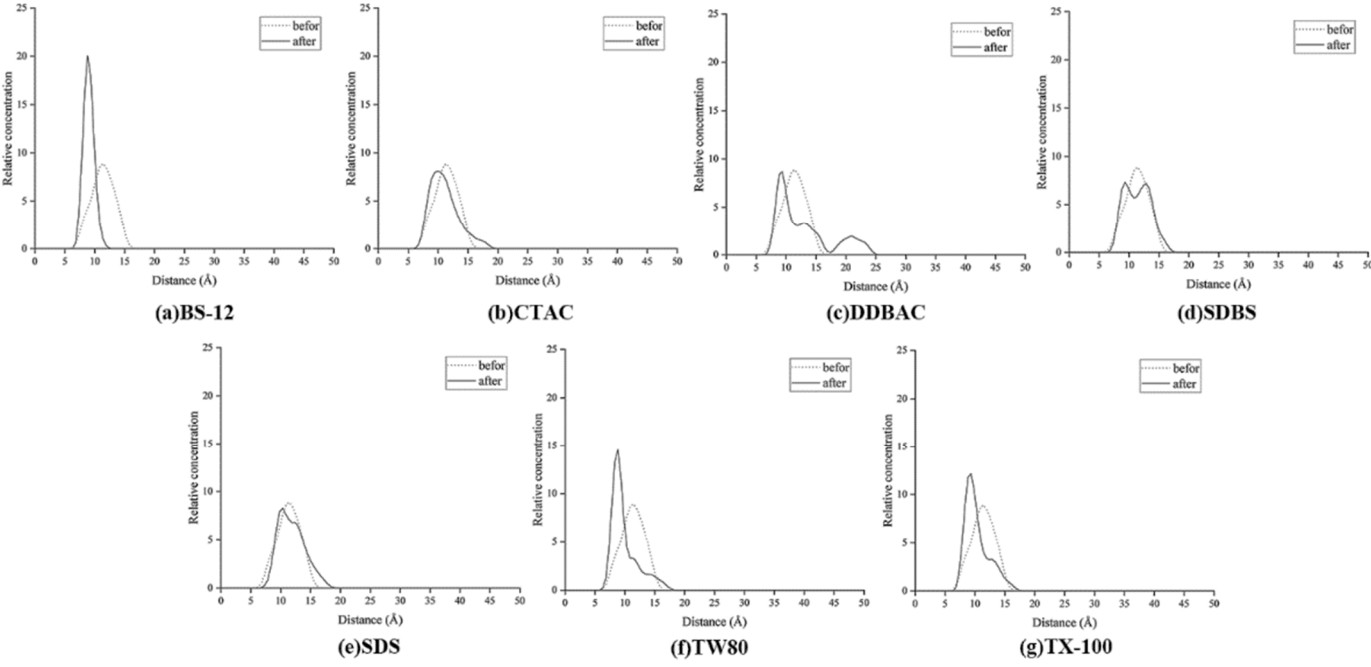

**Figure 8.** Concentration distribution of pyrene adsorbed on kaolinite surface under a low concentration of surfactant.

### 3.2. Desorption Test Results' Interaction between Collector and Kaolinite Surface

The results of pyrene desorption on the surface of kaolinite are shown in Figure 9. From Figure 9, it can be seen that with the change of the amount of surfactant added, there was a maximum pyrene removal rate for different types of surfactants. When the dosage of cationic surfactants DDBAC and CTAC was low, they had a good pyrene removal rate, but with the increase of dosage, the pyrene removal rate significantly decreased. When the amount of anionic surfactants SDS and SDBS added was small, they also had a certain pyrene removal rate. With an increase of the amount added, the pyrene removal rate of SDS gradually increased, and the pyrene removal rate of SDBS showed a trend of first increasing and then decreasing. The nonionic surfactants TX-100 and TW80 showed a maximum pyrene removal rate and then showed a decreasing trend with increasing addition. The pyrene removal rate of the amphoteric surfactant BS-12 showed a trend of first increasing and then decreasing with the increase of the amount added. When the amounts of SDS and DDBAC added were 25 mL and 10 mL, respectively, the removal rate of pyrene exceeded 70%, reaching 72.86% and 70.32%, respectively. These results indicate that the type and amount of surfactants have impacts on the pyrene removal rate.

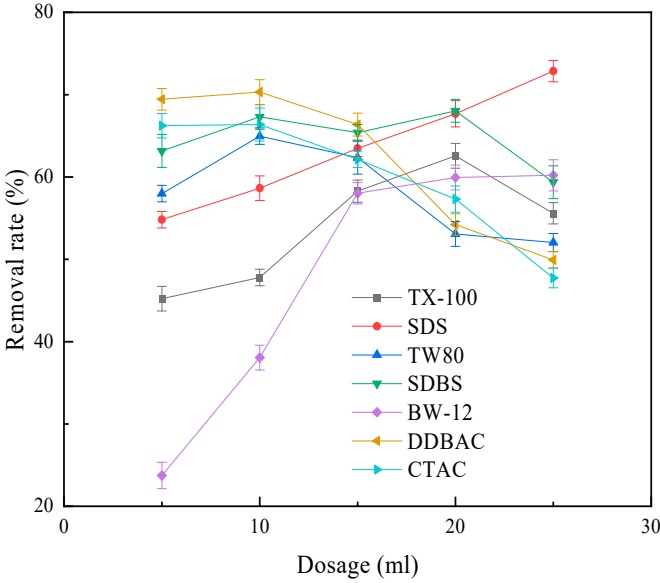

**Figure 9.** Results of pyrene desorption on the surface of kaolinite.

Cationic surfactants had a high removal rate of pyrene at low concentrations, but the removal rate gradually decreased with increasing concentration, which was consistent with molecular dynamics simulation results. The pyrene removal rate of anionic surfactant SDS gradually increased with the increase of concentration, while the pyrene removal rate of SDBS showed a trend of first increasing and then decreasing with the increase of concentration, which was different from the molecular simulation results. The possible reason is that the desorption test of SDS did not reach the maximum removal rate, so there was no downward trend. TRAN et al. [22] found through research that soil colloidal particles typically carry negative charges, indicating that cationic and anionic surfactants can bind to contaminated soil through ion exchange and ion matching. However, anionic surfactants show better removal results for pollutants in soil. SDS has been widely used due to its high efficiency, strong biodegradability, and low toxicity. Liu et al. [23] used SDS and CTAB to wash dioxins in incineration fly ash and found that CTAB had a stronger removal effect than SDS. At a certain concentration, the removal rate of dioxins increased with the increase of surfactants. However, when the concentration of surfactants is too high, it may lead to a decrease in the removal rate of dioxins. Rohi et al. [24] found through

experiments that cationic surfactants have a certain solubilization effect on polycyclic aromatic hydrocarbons.

The pyrene removal rate of nonionic surfactants showed a trend of first increasing and then decreasing, which was consistent with the results of molecular dynamics simulation. There were some differences between the results of the molecular dynamics simulation and the results of the amphoteric surfactant BS-12. This is because in the desorption test there was far more water than kaolinite. BS-12 is an amphoteric surfactant with two polarities, so BS-12 molecules first adsorb each other to form a self-agglomeration. When BS-12 reaches a certain concentration, BS-12 molecules will fully interact with a kaolinite pyrene system, thus presenting the results of molecular dynamics simulation.

## 4. Conclusions

In this study, molecular dynamics simulation and desorption tests were used to study the desorption of pyrene in kaolinite by different kinds of surfactants, and the following conclusions were drawn. Molecular dynamics simulation showed that pyrene was more easily adsorbed on the 001 (-) side of kaolinite. SDBS, SDS, TW80, and TX-100 had strong interactions with pyrene, encapsulating pyrene molecules in aggregates. The desorption of pyrene molecules will be inhibited in the presence of BS-12, TW80, and TX-100, while the desorption process can be promoted by using CTAC, DDBAC, SDBS, and SDS as soil remediation agents. When the concentration of SDS was 0.014 mol/L, the elution rate of pyrene reached 72.86%. The molecular dynamics simulation results are similar to the desorption test results, verifying the reliability of molecular dynamics simulation. These research results provide theoretical support for the selection of surfactants in the remediation process of pyrene-contaminated soil.

**Author Contributions:** Conceptualization, L.S.; methodology, L.S.; software, L.S.; validation, L.S., J.G. and Y.L.; formal analysis, J.G.; investigation, J.G.; resources, Y.L.; data curation, Y.L.; writing—original draft preparation, L.S.; writing—review and editing, E.Q.; visualization, E.Q.; supervision, E.Q.; project administration, E.Q.; funding acquisition, L.S. All authors have read and agreed to the published version of the manuscript.

**Funding:** This research was funded by the Natural Science Foundation of China (52104241); China Postdoctoral Science Foundation (2019M652163); Anhui Postdoctoral Science Foundation (2019B338).

**Informed Consent Statement:** Not applicable.

**Data Availability Statement:** The data relied on in this study are available in the article.

**Conflicts of Interest:** The authors declare no conflict of interest. The funders had no role in the design of the study; in the collection, analyses, or interpretation of data; in the writing of the manuscript, or in the decision to publish the results.

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
