# Peer review of "Study on the Remediation of Pyrene-Contaminated Soil with Surfactants and their Mechanisms"

_processes, doi:10.3390/pr11072199_

Round 1

Reviewer 1 Report

In this work effects and mechanism of different surfactants on the elution efficiency of pyrene in soil are studied. The effect of surfactants on the adsorption behavior of pyrene on kaolinite surface was studied by molecular dynamics simulation and desorption test. The manuscript can be improved after addressing the following points

1.      Revise the structure of the abstract and also state the novelty of the work.

2.      Correct the typos errors including unit correction such as “mol/l” should be “mol/L”. (Page 1, line 21)  

3.      Do not write the word/phrase and its abbreviation together again and again throughout the manuscript. See Page 2 (line 36 and 41). Abbreviate the word first and then use its abbreviation throughout the manuscript.

4.      Revise the introduction with latest references instead of old references of 19th century such as 1979.

5.      Use proper symbols and spaces between quantity and units throughout the manuscript. See page (line 93-98)

6.      Improve the quality of all Figures with correct ligand units.

7.      Some results in Figure 7 deviate from the simulated results. why?

8.      What is the significance of this study for practical applications?

Improve quality of writing by removing typos errors.  

Reviewer 2 Report

The authors study the adsorption and desorption of pyrene on kaolinite in the presence of various surfactants. The study has two components: a structural modelling component and a more experimental analysis of the pyrene removal from the kaolinite surface.

To be honest, I am not that sure what to think of the manuscript. I think the study is important, but I suspect that this is a case of the authors not explaining either the methods or their results in sufficient detail. Therefore, it is difficult for me to make an assessment on this.

For example, I think the main results are contained within figures 3,4 and 7. It is not clear what the models in figures 3 and 4 are really showing. In particular, it is not clear how pyrene is close to or far from away the surface of kaolinite (which the authors discuss), based on these images. This makes the authors’ claims somewhat weak. Also, what is the relevance of using 5 and 9 molecules in the analysis?

However, I am more interested (and somewhat concerned) with the desorption test results. My main concern is with sample replicates. This is a concern because the authors spend section 3.2 talking about the trends observed in these results. However, it is not clear whether there is actually a trend (in Figure 7), or whether the increases and decreases with various surfactants are just the result of the normal analytical error that occurs in all experiments. That is, with only 1 sample point, it is difficult to say with any confidence that there is any trend. I think this is the most concerning aspect of the manuscript.

Overall, I think this requires major revisions (if I am being generous), with the emphasis being on clarifying the experimental design and providing more detail with regards to the data interpretation. Please see my comments below.

Introduction:

Somewhere in the introduction I think it should be made clear why kaolinite might be a substrate for pyrene adsorption. I think it is important to clarify this because to be honest, I would suspect that the reason why PAHs are found in large amounts in soils is because of their adsorption to soil organic matter and not so much because of the mineral content. My guess is that a mineral like kaolinite is not a good substrate for PAH adsorption, given the polar functionalities present (though I could be wrong and if I am, please clarify). This is why I have always been suspicious of PAH-mineral surface adsorption studies. My guess is that if there was any possibility of interaction between PAH and mineral surface, it would be through the π-cation interactions. In any case, please address this. Why kaolinite?

Section 2.1. There is a table here that is not referenced in the text or explained. What is this for? Table 1 (which you reference in line 113 is actually found in line 121).

Line 98: This seems odd, so I would just like confirmation on this. So, pyrene from 9 g of kaolinite was extracted with only 1 mL of dichloromethane? My reflexive response is that this seems like a very small volume to use to extract from 9 g of mineral, but perhaps I am wrong. Please check this.

Section 2.4.1. The figure here needs a caption

Line 126: The idea that pyrene is adsorbed onto the 001 side is mentioned in both the abstract and the conclusion. However, it is not clear why this is relevant. Can the authors explain?

Line 186: You mention π-π stacking here, but the discussion is in the context of amphoteric surfactants. The amphoteric surfactant that is used in this study (BS-12) does not contain an aromatic group, as far as I know. Is this correct? If there is no aromatic group, I am not sure how the stacking occurs.

Line 198: In general, if the word “significant” is going to be used, it implies that some type of statistical test was done. Was there a statistical test done? If there were no sample replicates, this would be difficult…

There are some issues with missing words. For example in the abstract, line 10, soil is the main aggregation site for PAHs. The word "site", or something like that is missing. There are many instances of this in the manuscript. I think it just requires another round of proofreading to be honest.

Reviewer 3 Report

Review on “Study on the remediation of pyrene contaminated soil with surfactants and its mechanisms”

General: The authors studied the adsorption and desorption behaviors using various surfactants in contaminated soil samples. The introduction is appropriate, but the purpose of the study is not clearly presented, and the discussion is lacking. I do not understand why this study is important? If it is published in a Processes journal, many parts will have to be carefully revised.

Here are my comments,

1.       Line 71: What is the main purpose of the study? The purpose of the study, which is differentiated from other studies, must be clearly presented.

2.       Line 78: Tween 80 -> Tween 80 (TW80).

3.       Line 82: Table 1 includes the abbreviation and values for Mad (%), Aad (%), Vad (%), and Fcad (%). The authors must clearly state whate these abbreviations mean.

4.       Line 85: Why is Kaolinite pulverized below 125 μm.

5.       Line 91: There is no mention of Table 2 in the manuscript. If the authors do not need in this table, it is also a way to include it as a supplemenatary files.

6.       Line 112-113: The figure in page 3 has no caption. The authors should check the entire table and figure number carefully.

7.       Line 128: In figure 1, befor -> before. Do 001 and 001 (-) sides mean before and after respectively? The expression in this figure should be clear.

8.       Line 193-227: The authors are presenting simple results. It is necessary to enrich the discussion by citing related studies. Are there any other studies using SDS and DDBAC? In particular, further discussion should be included on whether SDS and DDBAC increase the removal rate at different dosage.

9.       Line 291-295: Where are 16 and 17 of the reference cited in the manuscript?

10.   Line 228: The conclusion should be rewritten.

11.   Reference section: The journal name of the cited paper should be expressed in abbreviations.

Round 2

Reviewer 1 Report

Accepted

Author Response

Thank you very much.

Reviewer 2 Report

Lines 80-98: This is where the authors address the previous comment about justifying kaolinite use. Based on what I have read in that paragraph, there does not seem to be any concrete idea of how pyrene is actually interacting with the kaolinite. Which is fine.  I was looking to see if the authors would address this. In lieu of this, the authors have just cited literature indicating that pyrene has some kind of interaction with minerals. This might be good enough. However, I would say that the paragraph is out of place and should be merged with the paragraph where PAHs are discussed (lines 51-60).

Line 85: Also, I would remove the Lambert reference. This is extremely misleading. Lambert is talking about the covalent bonds that may form from (for example) the carboxylic groups on an organic molecule and the mineral surface. Pyrene will not do that. In the context of your manuscript, this particular point you are making with respect to Lambert’s paper is irrelevant.

Lines 158-165: Previously the authors were asked to address the ambiguity of the extraction method. In their response they indicate that presumably 100 mL of dichloromethane is used in the extraction, with ultimately 1 mL being used for drying. If this is correct, then why is this not fixed in the manuscript? This type of information needs to be clear in the manuscript and not just assumed. I am becoming frustrated reading the author responses.

Lines 254-260: Ok, so in this case then what is the non ionic surfactant that is forming the π-π with the pyrene? I am asking for some specifics here. My original confusion was because I am not sure what particular surfactant you are talking about in the context of your research paper.

The mistakes are not so much grammatical, but stylistic. I would run this by a native English speaker to fix this.

Reviewer 3 Report

I am satisfied with the revised manuscript because many of the reviewers' suggestions have been corrected.

Author Response

Thank you very much.